# Liquiritin Suppresses Intracellular and Secreted MUC5AC and MUC5B in Human Airway Epithelial Cells

**DOI:** 10.3390/ijms26168076

**Published:** 2025-08-21

**Authors:** Ryoma Yoshio, Jun Iwashita

**Affiliations:** Faculty of Bioresource Sciences, Akita Prefectural University, Akita 010-0195, Japan; b23e040@akita-pu.ac.jp

**Keywords:** MUC5AC, MUC5B, asthma, liquiritin, flavonoid, mucin

## Abstract

The human airway surface is covered by a mucus layer composed primarily of the mucins MUC5AC and MUC5B. Excessive mucin production and secretion by airway epithelial cells in patients with asthma result in airway obstruction and worsened asthma symptoms. This study investigated the effects of liquiritin, a widely used flavonoid, on intracellular and secreted MUC5AC and MUC5B levels in the NCI-H292 human airway epithelial cell line. Liquiritin treatment suppressed both mucin types in a dose-dependent manner, accompanied by decreased activity of extracellular signal-regulated kinase (ERK) and p38. The effect of liquiritin was further examined in cells stimulated with phorbol 12-myristate 13-acetate (PMA) to induce excessive mucin production and secretion. Liquiritin dose-dependently reduced PMA-induced increases in intracellular and secreted MUC5AC and MUC5B levels as well as PMA-induced ERK and p38 activity. Overall, these results suggest that liquiritin reduces intracellular and secreted MUC5AC and MUC5B levels by suppressing the ERK and/or p38 signaling pathway.

## 1. Introduction

The surface of the healthy human lung and trachea is covered with a mucin-based mucosal layer, which plays a crucial role in biological defense against foreign substances [1,2]. Mucins are large glycoproteins consisting of a core protein to which numerous clustered sugar side chains are attached. In humans, at least 20 mucin subtypes are known and are classified as either membrane-bound or secretory types [3]. Membrane-bound mucins include MUC1, MUC3, and MUC4, whereas secretory mucins include MUC2, MUC5B, and MUC5AC. Airway epithelial homeostasis is maintained by two major gel-forming mucins, MUC5B and MUC5AC, which are essential for respiratory function. MUC5B predominates in the normal respiratory tract and is vital for host defense [4,5]. In contrast, MUC5AC is overproduced (by 40–200-fold) in the airways of patients with bronchial asthma. Airway mucus hypersecretion is a key pathophysiological feature of inflamed airways in asthma and impairs mucociliary transport [6,7]. In asthmatic airways, increased MUC5AC secretion leads to airway wall thickening and the formation of MUC5AC-rich mucus plugs, which contribute to airflow obstruction [1,2,8,9,10,11].

Various proinflammatory cytokines (e.g., IL-4, IL-6, and TNF-α) and cell adhesion molecules (e.g., cadherins and integrins) stimulate MUC5AC secretion. These factors regulate MUC5AC expression and contribute to mucus hypersecretion in lung epithelial cells [12,13,14,15,16]. Signaling initiated by these stimuli activates the epidermal growth factor receptor, extracellular signal-regulated kinase (ERK), and protein kinase C (PKC) [2,17]. Additionally, the p38 mitogen-activated protein kinase (MAPK) pathway is activated by various stimuli, such as the proinflammatory cytokine IL-6 [18,19,20]. This leads to activation of the transcription factor nuclear factor-κB, resulting in increased MUC5AC secretion [13,21,22]. Moreover, activation of the MEK/ERK and p38 pathways further promotes MUC5AC production. Because MUC5AC oversecretion exacerbates asthma symptoms, effective asthma therapies should target the reduction of MUC5AC production and secretion. Developing such therapies requires a thorough understanding of the regulatory mechanisms underlying mucin expression. Although aerosolized administration of steroid drugs used to treat respiratory diseases such as asthma reduce airway inflammation and prevent airway narrowing, they are associated with side effects, including immunosuppression of the mucosal surfaces. Therefore, there is increasing interest in long-term treatments using natural compounds, such as plant-derived ingredients, which are associated with fewer side effects.

Plant-derived compounds have been reported to be effective against various diseases. For example, prunin, a flavonoid glycoside presents in many plants and fruits, exhibits potential anticancer effects by modulating key signaling pathways [23]. Licorice, a genus of plants in the family Amaryllidaceae, is a processed herb extracted from the root of Spanish licorice (*Glycyrrhiza glabra*). It is widely used in foods, beverages, and traditional Chinese herbal medicine, where it has been reported to possess anti-inflammatory properties [24]. The root of licorice contains over 400 identified compounds, including triterpenoid saponins, flavonoids [25,26], phenolics, and other bioactive compounds [27]. Flavonoids, in particular, have demonstrated therapeutic potential against various diseases, including respiratory conditions [28,29,30]. Over 300 flavonoids have been extracted from licorice, including liquiritin and glycyrrhizin. Glycyrrhizin has been shown to inhibit mucus hyperproduction and MUC5AC gene transcription both in vivo and in vitro [31]. Liquiritin, a major flavonoid constituent of licorice, is used as a therapeutic agent for cough suppression [20,21,22]. It was originally identified through flavonoid screening for its neuroprotective effects via inhibition of TRPV1 and TRPA1 channels [32]. Liquiritin also exerts anti-inflammatory effects in human cells by inhibiting the expression of IL-1β and IL-6 [33]. Furthermore, it reduces the levels of proinflammatory cytokines IL-1β, IL-6, and TNF-α in mice, which are cytokines known to stimulate MUC5AC secretion [34].

Given its reported anti-inflammatory and anti-allergic effects, liquiritin may help alleviate asthma symptoms associated with inflammation and allergies. Although mucin production has previously been linked to key signaling pathways such as ERK and p38, its specific effects on these pathways in airway epithelial cells have not been fully elucidated. This study focused on the mechanisms underlying liquiritin’s effects on the production and secretion of MUC5AC and MUC5B. In this report, we found that liquiritin reduced both intracellular and secreted MUC5AC and MUC5B levels, accompanied by suppression of ERK and p38 activity. These findings indicate that liquiritin may be a promising therapeutic agent for suppressing the production and secretion of MUC5AC and MUC5B in the airway of asthmatic patients.

## 2. Results

### 2.1. Effect of Liquiritin on Intracellular and Secreted MUC5AC and MUC5B Levels

First, we assessed the cytotoxicity of liquiritin (50, 100, 150, 200, 300 and 400 µM for 72 h) in both phorbol 12-myristate 13-acetate (PMA)-untreated (Figure 1A) and PMA-treated (Figure 1B) NCI-H292 cells. Treatment with liquiritin at concentrations up to 300 µM showed no significant cytotoxicity over a 72 h period. Subsequent analysis was performed at liquiritin concentrations up to 200 µM, where no cytotoxicity was observed. We then evaluated the effect of liquiritin (50, 100, 150 and 200 µM for 72 h) on intracellular and secreted levels of MUC5AC and MUC5B. Dot blot analysis revealed that intracellular MUC5AC levels were dose-dependently suppressed by liquiritin at concentrations of 50–200 µM in PMA-untreated NCI-H292 cells (Figure 2A). Similarly, secreted MUC5AC levels in the culture medium were significantly decreased by liquiritin at 150–200 µM (Figure 2B). Additionally, treatment with 100 µM liquiritin for 72 h reduced both intracellular (Figure 3A) and secreted (Figure 3B) MUC5B levels to approximately 50% of control values.

### 2.2. Effects of Liquiritin on the Phosphorylation of ERK and p38 in NCI-H292 Cells

Since the addition of liquiritin decreased intracellular and secreted MUC5AC and MUC5B, we next analyzed its regulatory mechanism. Phosphorylated (activated) ERK and p38 are known to promote the expression of MUC5AC and MUC5B, respectively. Therefore, in PMA-untreated NCI-H292 cells, we analyzed the effect of liquiritin (50, 100, 150, and 200 µM for 72 h) on the basal phosphorylation levels of ERK and p38 using Western blot with anti-phospho-ERK and anti-phospho-p38 antibodies. α-Tubulin was used as a loading control. Band intensities were quantified, and the ratio of phosphorylated ERK to total ERK was calculated and plotted. The results demonstrated that liquiritin reduced ERK phosphorylation after 72 h of treatment (Figure 4). Additionally, liquiritin decreased p38 phosphorylation in a dose-dependent manner over the same period (Figure 5).

### 2.3. Effect of Liquiritin on Intracellular and Secreted MUC5AC and MUC5B Levels in PMA-Stimulated Cells

Since the addition of liquiritin decreased normal level of intracellular and secreted MUC5AC and MUC5B expression, we next analyzed the effect of liquiritin on stimulated and overexpressed of MUC5AC and MUC5B. PMA treatment significantly increased both intracellular and secreted levels of MUC5AC and MUC5B in airway epithelial cells, mimicking the inflammatory state observed in human asthmatic lungs. To evaluate the potential inhibitory effect of liquiritin under inflammatory conditions, we treated PMA-stimulated NCI-H292 cells with liquiritin (50, 100, 150, and 200 µM for 72 h) and assessed MUC5AC and MUC5B levels via dot blotting. Intracellular MUC5AC levels were dose-dependently suppressed by liquiritin, with significant inhibition observed at 100 µM (Figure 6A). Similarly, secreted MUC5AC levels in the culture medium were significantly reduced at liquiritin concentrations ranging from 50 to 200 µM (Figure 6B). Treatment with 100 µM liquiritin for 72 h also decreased both intracellular (Figure 7A) and secreted (Figure 7B) MUC5B levels in PMA-stimulated cells.

### 2.4. Effects of Liquiritin on the Phosphorylation of ERK and p38 in PMA-Stimulated Cells

PMA activates ERK and p38 through phosphorylation, thereby inducing the over-production of MUC5AC and MUC5B. To assess the regulatory effect of liquiritin under these conditions, we treated PMA-stimulated NCI-H292 cells with liquiritin (50, 100, 150, and 200 µM for 72 h) and analyzed the phosphorylation status of ERK and p38 via west-ern blotting using anti-phospho-ERK and anti-phospho-p38 antibodies. α-Tubulin was used as a loading control. Band intensities were quantified, and the ratios of phosphorylated ERK to total ERK and phosphorylated p38 to total p38 were calculated and graphed. The results showed that liquiritin significantly reduced ERK phosphorylation in PMA-stimulated cells after 72 h of treatment (Figure 8). Furthermore, liquiritin decreased p38 phosphorylation in a dose-dependent manner under the same conditions (Figure 9).

## 3. Discussion

Liquiritin, commonly used as a cough suppressant, has been reported to reduce the mRNA expression levels of inflammatory cytokines such as IL-1β, IL-6, and TNF-α in mice, which are factors known to promote MUC5AC secretion in inflamed airways [33,34]. Therefore, this study focused on evaluating the effect of liquiritin on MUC5AC and MUC5B production and secretion and investigated the underlying signaling pathways involved in mucin regulation in vitro.

MUC5AC and MUC5B share domain similarities but differ in their multimeric organization, reflecting distinct structural characteristics [35]. MUC5AC tends to form branched networks, whereas MUC5B forms linear networks with occasional branching [35]. In inflamed airways, MUC5B contributes to honeycomb-like structures and plays a role in mucociliary transport, whereas MUC5AC is associated with mucus plug formation and airway hyperresponsiveness [36,37]. Although MUC5AC levels increase disproportionately and this increase is more closely linked to pathological airway changes in inflammatory diseases, both MUC5B and MUC5AC levels are significantly elevated in in-flamed airways [38]. Therefore, regulating MUC5B expression is as important as regulating MUC5AC for effective asthma treatment.

Our results suggest that the suppression of MUC5AC and MUC5B by liquiritin was mediated via inhibition of the constitutive or PMA-activated ERK and/or p38 signaling pathways. The secretion of MUC5AC and MUC5B is regulated by multiple mechanisms [39]. To determine whether liquiritin affects intracellular synthesis or the secretory process, we separately analyzed intracellular and secreted levels of MUC5AC and MUC5B. Liquiritin treatment similarly reduced both intracellular and secreted levels of these mucins, suggesting that its effect is not limited to inhibiting secretion but may also involve suppression of mucin synthesis.

Next, we used PMA-stimulated NCI-H292 cells, which overproduce mucin, to mimic the inflammatory state of human asthmatic lung cells. PMA is a widely used activator of PKC that induces mucin secretion. It has been shown to stimulate MUC5AC secretion in human bronchial epithelial cells via PKC activation [40,41] and to upregulate both MUC5AC and MUC5B through the MAPK pathway [40,42]. Previous studies have investigated the effects of EGF or PMA stimulation on mucin expression in airway epithelial cells [43,44]. Emodin, a natural trihydroxyanthraquinone compound found in the roots and barks of plants, inhibits the expression of MUC5AC mRNA and protein by 30% to 50% via suppressing the phosphorylation of EGF receptor, ERK and p38 [43]. Pyronaridine, an antimalarial agent, suppresses PMA induced overexpression of MUC5AC mRNA to control level in NCI-H292 cells [44]. In this study, we examined the effect of liquiritin on intracellular and secreted MUC5AC and MUC5B levels in PMA-stimulated cells. Our results showed a significant increase in both intracellular and secreted MUC5AC and MUC5B levels following PMA stimulation, which was attenuated to control level after 72 h of liquiritin treatment (Figure 6 and Figure 7). It was observed that liquiritin also had the same level of inhibitory effect on MUC5AC expression as emodin or pyronaridine. These findings indicate that liquiritin suppresses MUC5AC and MUC5B levels in both PMA-stimulated and unstimulated cells, suggesting that it inhibits both constitutive and inducible mucin production and secretion. These reductions were accompanied by decreased phosphorylation of ERK and p38 (Figure 8 and Figure 9), indicating that the inhibitory effect of liquiritin on MUC5AC and MUC5B in PMA-stimulated cells is mediated, at least in part, through suppression of the ERK and/or p38 signaling pathways.

Our results suggest that liquiritin may help mitigate asthma symptoms by inhibiting both intracellular and secreted levels of MUC5AC and MUC5B through suppression of ERK and p38 signaling. However, the differential effects observed on ERK and p38 indicate that additional signaling molecules may also play a significant role in the suppression of MUC5AC and MUC5B production and secretion. In particular, c-Jun N-terminal kinase (JNK), which is known to activate MUC5AC and MUC5B expression, may also be inhibited by liquiritin [45,46]. We plan to investigate the effect of liquiritin on JNK activity in future studies.

We believe that our finding of the new effect of plant-derived small molecules on mucin expression is valuable. However, there are limits to in vitro models using cultured cell line to analyze biological significance. Therefore, additionally, we aim to further evaluate the effects of liquiritin on both constitutive and inducible MUC5AC secretion in vivo using asthmatic mouse models. We plan to try treatment using airway sprays or intravenous administration with asthmatic model mice in future studies.

## 4. Materials and Methods

### 4.1. Experimental Design

We aimed to explore the potential of long-term treatments using natural ingredients, specifically plant-derived flavonoids. We hypothesized that liquiritin, a flavonoid with known anti-inflammatory properties, may suppress the excessive production and secretion of the mucins MUC5AC and MUC5B, which contribute to airway inflammation in diseases such as asthma. To test this hypothesis, liquiritin was added to the human air-way epithelial cell line NCI-H292, which naturally produces and secretes MUC5AC and MUC5B. Additionally, PMA-treated NCI-H292 cells were used to model the inflammatory state of asthmatic airways, as PMA induces mucin overproduction. Following treatment with liquiritin, both intracellular and secreted levels of MUC5AC and MUC5B were measured using the dot blot method with specific antibodies. Furthermore, the activity of the ERK and p38 signaling pathways, which are known regulators of mucin production, was assessed via Western blot analysis to elucidate the underlying molecular mechanisms.

### 4.2. Cell Culture

The human lung cancer cell line NCI-H292 was purchased from the American Type Culture Collection (Gaithersburg, MD, USA). NCI-H292 cells were cultured in RPMI-1640 medium (Sig-ma-Aldrich, Tokyo, Japan) supplemented with 10% fetal bovine serum (FBS; Cansera In-ternational, Etobicoke, ON, Canada), 100 units/mL penicillin (Gibco Oriental, Tokyo, Japan), and 100 μg/mL streptomycin (Gibco Oriental, Tokyo, Japan) in a 5% CO_2_ incubator. Adherent cells were subcultured every 3–4 days using a trypsin-EDTA solution (Gibco Ori-ental, Tokyo, Japan). Cells were seeded in culture plates, including 96-well plates (MS-8096F, Sumilon, Tokyo, Japan). Liquiritin (Fujifilm Wako Pure Chemical Corporation, Osaka, Japan)) and PMA (GTX24819, Funakoshi, Tokyo, Japan) were dissolved in ethanol (Wako, Tokyo, Japan) before use.

### 4.3. Treatment of Cells with PMA and Liquiritin

NCI-H292 cells (1 × 10^4^ cells in 0.1 mL) were seeded and cultured to confluence in 96-well plates at 37 °C. For serum deprivation, the confluent cells were incubated in RPMI-1640 medium containing 0.2% FBS for 24 h. After deprivation, the cells were pretreated with liquiritin ((Fujifilm Wako Pure Chemical Corporation, Osaka, Japan) in RPMI 1640 with 0.2% FBS at the indicated concentrations (50, 100, 150, and 200 μM) for 30 min. PMA was then added to the culture medium (RPMI 1640 with 0.2% FBS) at a final concentration of 20 μM and cultured for 72 h and sampled. In the PMA-untreated group, cells were treated with liquiritin alone at the indicated concentrations for 72 h before sampling.

### 4.4. Cell Proliferation Assay

Cell proliferation was assessed using the Cell Counting Kit-8 (Dojindo, Kumamoto, Japan). NCI-H292 cells (1 × 10^4^ cells in 0.1 mL) were cultured in 96-well plates for 30 h at 37 °C. Afterward, 0.01 mL of the kit reagent was added to each well, and the plate was incubated for an additional 2 h at 37 °C. Cell growth was evaluated by measuring absorbance at 450 nm using a Model 550 microplate reader (Bio-Rad, Tokyo, Japan).

### 4.5. MUC5AC and MUC5B Protein Level Measurement Using the Dot Blot Method

Samples collected from cells or culture medium were diluted 1:500 in Tris-buffered saline (20 mM Tris, pH 7.4, 150 mM NaCl) containing 0.1% sodium dodecyl sulfate (SDS). The diluted solution (10 μL) was blotted onto an Immobilon membrane (Millipore, Temecula, CA, USA) using a Dot Blot Hybridization Manifold (48-well, SCIE-PLAS, Cambridge, UK). For measurement of cellular MUC5AC and MUC5B levels, 5 μL of the cell lysate was directly applied to the membrane. The membrane was then blocked with Western blot blocking buffer (T7131A, Takara, Tokyo, Japan) in Tris-buffered saline with 0.1% Tween 20 (TBS-T) for 12 h at 4 °C. After blocking, the membrane was incubated for 1 h with either a mouse anti-MUC5AC antibody (MS145-P1, 1:2000 in Western blot blocking buffer; Neomarkers, Fremont, CA, USA) or a mouse anti-MUC5B antibody (ab77995, 1:2000 in 4% skim milk; Abcam, Tokyo, Japan). The membrane was washed five times with TBS-T (5 min each) and then incubated for 1 h with a rabbit anti-mouse IgG (H + L) secondary anti-body (NA931V, 1:2000 in 4% skim milk; GE Healthcare, Buckinghamshire, UK). Following another five washes, the immunoreactive signals were detected using the Luminata Forte Western HRP substrate (WBLUF0500, Millipore) and visualized using a ChemiDoc image analyzer (Bio-Rad). The results were analyzed using Bio-Rad Image Lab Software version 5.0.

### 4.6. Immunoblot Detection of α-Tubulin, Phosphorylated ERK, Total ERK, Phosphorylated p38, and Total p38 Levels

For detecting cellular protein levels, cells were lysed in 1× Laemmli sample buffer (161-074, Bio-Rad, Tokyo, Japan). Proteins were separated via electrophoresis on a 10% SDS-PAGE gel using an IEP-1010 electrophoresis apparatus (AXEL, Tokyo, Japan) and then transferred to a nitrocellulose membrane (Hybond ECL, GE Healthcare) using a BE351 transfer apparatus (Bio Craft, Tokyo, Japan). The membrane was blocked in west-ern blot blocking buffer diluted in TBS-T for 12 h at 4 °C and subsequently incubated for 1 h with one of the following primary antibodies (all at 1:2000 dilution in blocking buffer): rabbit anti-α-tubulin (PM054, MBL, Tokyo, Japan), rabbit anti-phosphorylated (active) ERK1/2 (GTX24819, Funakoshi, Tokyo, Japan), rabbit anti-total ERK1/2 (V1141, Promega, Madison, WI, USA), rabbit anti-phosphorylated (active) p38 MAPK (pThr180/Tyr182) (GTX133460, GeneTex, Irvine, CA, USA), or rabbit anti-total p38 MAPK (#9212S, Cell Sig-naling Technology, Danvers, MA, USA). After incubation with the primary antibodies, the mem-brane was washed five times with TBS-T (5 min each) and incubated for 1 h with HRP-conjugated anti-rabbit IgG (W4011, Promega) at a 1:2000 dilution. After another five washes, immunoreactive bands were visualized using Luminata Forte Western HRP Sub-strate (Millipore) and a ChemiDoc image analyzer (Bio-Rad). The membrane was then stripped using Restore Western blot Stripping Buffer (21059, Thermo Fisher Scientific, Rockford, IL, USA) for 15 min at room temperature with shaking. After stripping, the membrane was washed five times with TBS-T (5 min each) and reblocked with Western blot blocking buffer for 12 h at 4 °C. The results were analyzed using Bio-Rad Image Lab Software version 5.0.

### 4.7. Statistical Analysis

Differences between experimental groups were analyzed using analysis of variance (ANOVA, Dunnett’s test) and two-tailed unpaired Student’s *t*-test. ANOVA was used for comparisons involving more than two groups. Both ANOVA and *t*-tests were performed, and *p*-values were calculated using Microsoft Excel (Office Professional Plus 2016). A *p*-value of <0.05 was considered statistically significant. All experiments were performed in triplicate, and representative results are presented. “ns” indicates a nonsignificant difference.

## 5. Conclusions

Exposure to liquiritin reduced both intracellular and secreted levels of MUC5AC and MUC5B in NCI-H292 cells, accompanied by suppressed ERK and p38 activity. In PMA-treated cells, liquiritin also decreased intracellular and secreted MUC5AC and MUC5B, accompanied by suppressed ERK and p38. These results suggest that liquiritin attenuates MUC5AC hypersecretion, at least in part, through inhibition of ERK and/or p38 signaling. Therefore, liquiritin may hold potential as a therapeutic agent for improving asthma treatment. Suppression of excessively secreted MUC5AC and MUC5B is expected to alleviate airway narrowing and asthma symptoms.

## Figures and Tables

**Figure 1 ijms-26-08076-f001:**
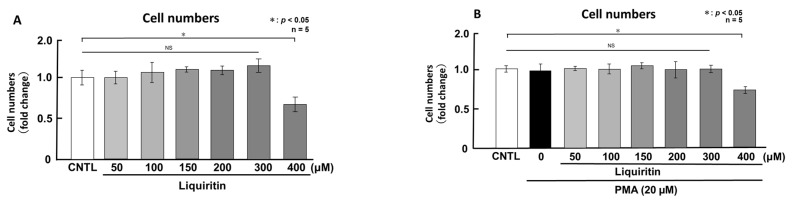
Effects of liquiritin on cell proliferation in PMA-treated and -untreated cells. PMA-treated (20 μM) and -untreated NCI-H292 cells were exposed to varying concentrations of liquiritin for 72 h. Cell proliferation was assessed using the Cell Counting Kit-8. (**A**) Cell proliferation levels of PMA-untreated cells. (**B**) Cell proliferation levels of PMA-treated cells. Graphs show the relative proliferation levels of liquiritin-treated cells compared with control cells (treated with vehicle alone). Fold changes were calculated as the ratio of treated to control values (mean ± standard deviation [SD], *n* = 5). Statistical significance was determined using Student’s *t*-test; *p* < 0.05 was considered significant. Representative results from three independent experiments are shown.

**Figure 2 ijms-26-08076-f002:**
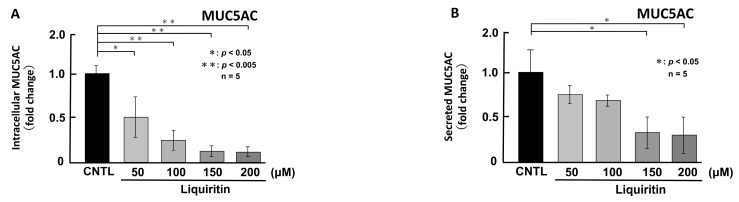
Effects of liquiritin on intracellular and secreted MUC5AC levels in PMA-untreated cells. NCI-H292 cells were treated with varying concentrations of liquiritin for 72 h. Intracellular and secreted MUC5AC levels were analyzed via dot blotting from cell lysates (**A**) and culture medium (**B**), respectively. Graphs show relative MUC5AC levels in liquiritin-treated cells compared with control cells (treated with vehicle alone). Fold changes were calculated based on the ratio between treated and control values (mean ± SD, *n* = 5). Statistical significance was determined using Student’s *t*-test; *p* < 0.05 was considered significant. Representative results from three independent experiments are shown.

**Figure 3 ijms-26-08076-f003:**
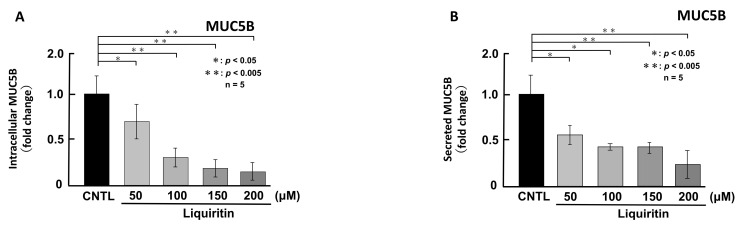
Effects of liquiritin on intracellular and secreted MUC5B levels in PMA-untreated cells. NCI-H292 cells were treated with varying concentrations of liquiritin for 72 h. Intracellular and secreted MUC5B levels were analyzed via dot blotting from cell lysates (**A**) and culture medium (**B**), respectively. Graphs show relative MUC5B levels in liquiritin-treated cells compared with control cells (treated with vehicle alone). Fold changes were calculated based on the ratio between treated and control values (mean ± SD, *n* = 5). Statistical significance was determined using Student’s *t*-test; *p* < 0.05 was considered significant. Representative results from three independent experiments are shown.

**Figure 4 ijms-26-08076-f004:**
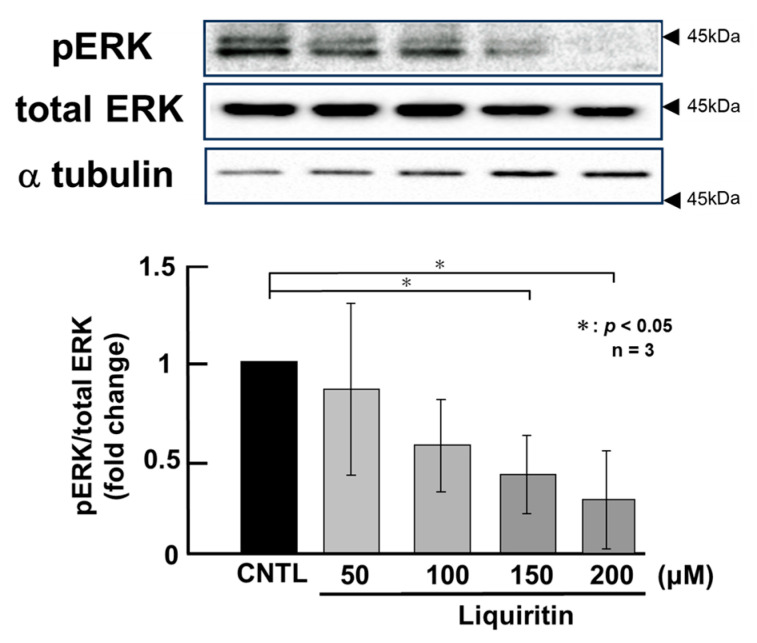
Effects of liquiritin on ERK phosphorylation in PMA-untreated cells. NCI-H292 cells were treated with varying concentrations of liquiritin for 72 h. Whole-cell extracts were collected and analyzed via Western blot to assess levels of phosphorylated ERK (pERK), total ERK, and α-tubulin (used as a loading control). Graphs show fold changes in phosphorylated protein levels relative to total protein levels (mean ± SD, *n* = 3). Statistical significance was determined using Student’s *t*-test; *p* < 0.05 was considered significant. Representative results from three independent experiments are shown. Black arrow heads indicate molecular weight (kDa).

**Figure 5 ijms-26-08076-f005:**
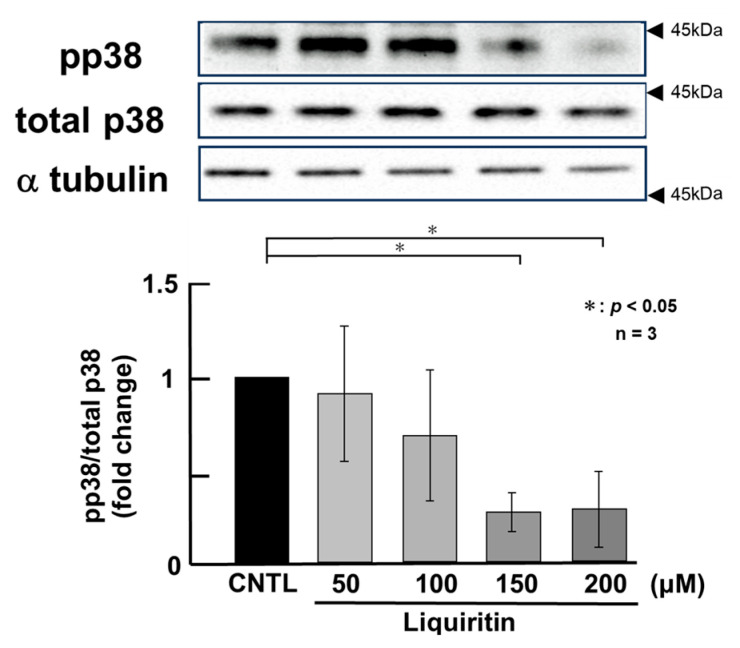
Effects of liquiritin on p38 phosphorylation in PMA-untreated cells. NCI-H292 cells were treated with varying concentrations of liquiritin for 72 h. Whole-cell extracts were collected and analyzed via Western blot to assess levels of phosphorylated p38 (pp38), total p38, and α-tubulin (used as a loading control). Graphs show fold changes in phosphorylated protein levels relative to total protein levels (mean ± SD, *n* = 3). Statistical significance was determined using Student’s *t*-test; *p* < 0.05 was considered significant. Representative results from three independent experiments are shown. Black arrow heads indicate molecular weight (kDa).

**Figure 6 ijms-26-08076-f006:**
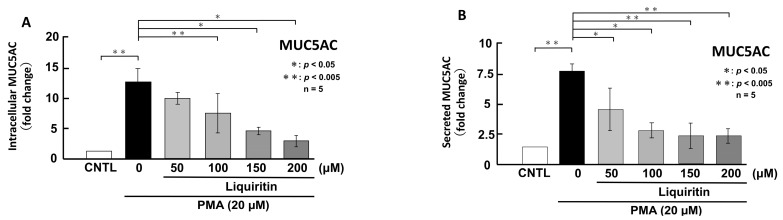
Effects of liquiritin on intracellular and secreted MUC5AC levels in PMA-treated cells. NCI-H292 cells were treated with 20 μM PMA and varying concentrations of liquiritin for 72 h. Intracellular and secreted MUC5AC levels were analyzed via dot blotting from cell lysates (**A**) and culture medium (**B**), respectively. Graphs show relative MUC5AC levels in liquiritin-treated cells compared with control cells (treated with vehicle alone). Fold changes were calculated based on the ratio between treated and control values (mean ± SD, *n* = 5). Statistical significance was determined using Student’s *t*-test; *p* < 0.05 was considered significant. Representative results from three independent experiments are shown.

**Figure 7 ijms-26-08076-f007:**
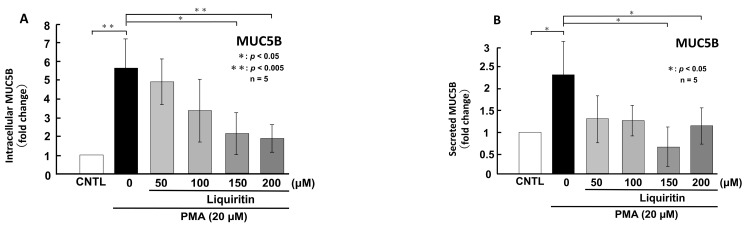
Effects of liquiritin on intracellular and secreted MUC5B levels in PMA-treated cells. NCI-H292 cells were treated with 20 μM PMA and varying concentrations of liquiritin for 72 h. Intracellular and secreted MUC5B levels were analyzed via dot blotting from cell lysates (**A**) and culture medium (**B**), respectively. Graphs show relative MUC5B levels in liquiritin-treated cells compared with control cells (treated with vehicle alone). Fold changes were calculated based on the ratio between treated and control values (mean ± SD, *n* = 5). Statistical significance was determined using Student’s *t*-test; *p* < 0.05 was considered significant. Representative results from three independent experiments are shown.

**Figure 8 ijms-26-08076-f008:**
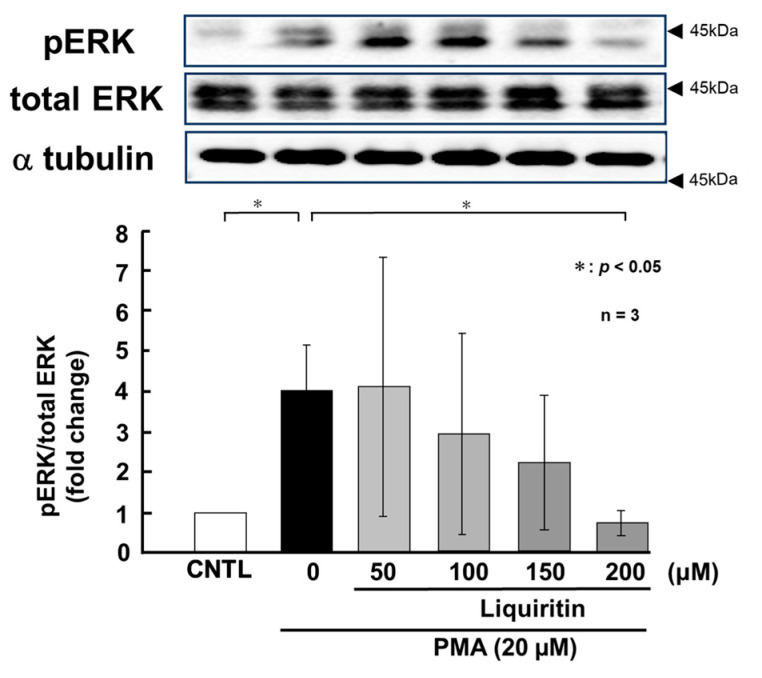
Effects of liquiritin on ERK phosphorylation in PMA-stimulated cells. NCI-H292 cells were treated with 20 μM PMA and varying concentrations of liquiritin for 72 h. Whole-cell extracts were collected and analyzed via Western blot to assess levels of phosphorylated ERK (pERK), total ERK, and α-tubulin (used as a loading control). Graphs show fold changes in phosphorylated protein levels relative to total protein levels (mean ± SD, *n* = 3). Statistical significance was determined using Student’s *t*-test; *p* < 0.05 was considered significant. Representative results from three independent experiments are shown. Black arrow heads indicate molecular weight (kDa).

**Figure 9 ijms-26-08076-f009:**
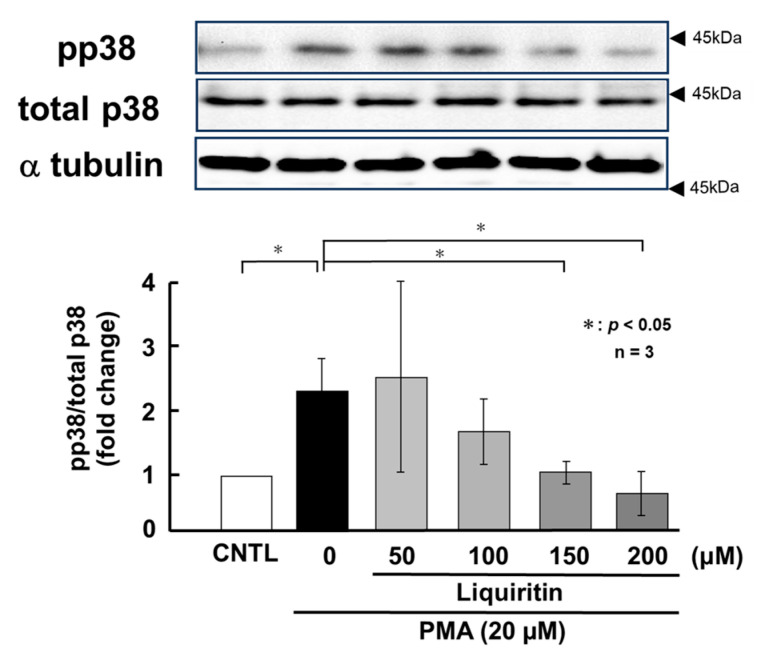
Effects of liquiritin on p38 phosphorylation in PMA-stimulated cells. NCI-H292 cells were treated with 20 μM PMA and varying concentrations of liquiritin for 72 h. Whole-cell extracts were collected and analyzed via Western blot to assess levels of phosphorylated p38 (pp38), total p38, and α-tubulin (used as a loading control). Graphs show fold changes in phosphorylated protein levels relative to total protein levels (mean ± SD, *n* = 3). Statistical significance was determined using Student’s *t*-test; *p* < 0.05 was considered significant. Representative results from three independent experiments are shown. Black arrow heads indicate molecular weight (kDa).

## Data Availability

The raw data supporting the conclusions of this article will be made available by the authors, without undue reservation. The original contributions presented in the study are included in the article. Further inquiries can be directed to the corresponding author.

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
