# Peer review of "Liquiritin Suppresses Intracellular and Secreted MUC5AC and MUC5B in Human Airway Epithelial Cells"

_ijms, 2025, doi:10.3390/ijms26168076_

Round 1
Reviewer 1 Report
Comments and Suggestions for Authors
This study examines how liquiritin affects MUC5AC and MUC5B mucin production in NCI-H292 human airway epithelial cells, both unstimulated and stimulated with PMA to simulate mucus overproduction in asthma. Liquiritin reduces the levels of intracellular and secreted MUC5AC and MUC5B in a dose-dependent manner, along with decreased phosphorylation of ERK and p38, indicating that it inhibits these pathways. In cells treated with PMA, liquiritin similarly reduces mucin overproduction and signaling activity. These results are important for asthma therapy, as excessive mucin contributes to airway blockage. Liquiritin's ability to regulate mucin through signaling pathways offers a promising new treatment strategy. However, significant revisions are required before publication.
Comments for authors
Comment 1. The introduction discusses mucin subtypes briefly. Include a review of 2023-2025 studies on flavonoids in airway diseases to justify liquiritin’s study.
Comment 2. What specific research gaps does this study address? Add details on the scientific questions or unresolved issues targeted.
Comment 3. The introduction lacks the recent literature on biomedical applications of plant-derived compounds. I recommend the inclusion of a very recent study in the introduction section:
J.N. Rana, S. Mumtaz, Prunin: An Emerging Anticancer Flavonoid, Int. J. Mol. Sci. 26 (2025). https://doi.org/10.3390/ijms26062678.
Comment 4. The cell proliferation assay results are presented, but the text does not specify the liquiritin concentrations tested.
Comment 5. In Western blot analysis, JNK should be tested along with ERK and P38.
Comment 6. Western blots for ERK and p38 phosphorylation omit molecular weight markers and full gels. Include these for band specificity and transparency.
Comment 7. The manuscript contains numerous grammatical errors. A thorough proofreading and editing process is essential to correct these errors, standardize terminology, and ensure clarity throughout the manuscript.
Looking forward to reviewing the revised version.
Author Response
Comments 1. The introduction discusses mucin subtypes briefly. Include a review of 2023-2025 studies on flavonoids in airway diseases to justify liquiritin’s study.
Response 1: Thank you for your helpful comments. References 28–30, which are review articles on flavonoids in airway diseases, have been, accordingly, added to the reference section.
Comments 2. What specific research gaps does this study address? Add details on the scientific questions or unresolved issues targeted.
Response 2: We agree with this comment. Regarding this issue, we have, accordingly, added the description to the Introduction section (line 48-55 and 75-78).
Comments 3. The introduction lacks the recent literature on biomedical applications of plant-derived compounds. I recommend the inclusion of a very recent study in the introduction section:
J.N. Rana, S. Mumtaz, Prunin: An Emerging Anticancer Flavonoid, Int. J. Mol. Sci. 26 (2025). https://doi.org/10.3390/ijms26062678.
Response 3: Thank you for your kind comment. We have added sentences referencing this recent study, which has been included as Reference 23 in the line 57-58 in the revised manuscript.
Comments 4. The cell proliferation assay results are presented, but the text does not specify the liquiritin concentrations tested.
Response 4: As suggested, we have added the liquiritin concentrations tested in the line 86-87 int the Results section for clarity.
Comments 5. In Western blot analysis, JNK should be tested along with ERK and P38.
Response 5: We agree with your point. We plan to prepare JNK antibodies and conduct further analyses. A corresponding sentence has been added to the lane 277-282 in the Discussion section.
Comments 6. Western blots for ERK and p38 phosphorylation omit molecular weight markers and full gels. Include these for band specificity and transparency.
Response 6: As requested, we have included molecular weight markers in the relevant Figure 4, 5, 8 and 9. Full gel images with molecular weight markers were posted separately as blots data which used in the figures.
Comments 7. The manuscript contains numerous grammatical errors. A thorough proofreading and editing process is essential to correct these errors, standardize terminology, and ensure clarity throughout the manuscript.
290-302
Response 7: As requested, the manuscript has been re-proofread by Enago to ensure accuracy and clarity in English.
Reviewer 2 Report
Comments and Suggestions for Authors
Yoshio et al. studied the effect of liquiritigenin found in licorice, on the secretion of two mucins, MUC5AC and MUC5B, using primary human airway epithelial cells (HAEPI). Excessive secretion of mucins has been shown to lead to airway obstruction and worsening asthma in patients with asthma. Study results showed that liquiritigenin was capable of dose-dependently inhibiting the levels of both intracellular and secretory mucins, even when cells were stimulated by phorbol ester (PMA) to secrete excessive mucins. Furthermore, the ability of liquiritigenin to inhibit the mucin secretion has been shown to correlate with a decrease in the activation of the ERK, p38 signaling pathways. These results suggest that liquiritigenin may inhibit excessive secretions of mucins through its regulation of the ERK and p38 signaling pathways and is a possible new avenue for asthma treatment.
This study's main limitations or improvements, in my view, are the study design and scope, as follows:
- Limitations of in vitro study: The study was carried out in vitro in the NCI-H292 human airway epithelial cell line. Even though this cell model can be used for mimicking the inflammatory state of human asthmatic lung cells (using PMA stimulation) and has yielded promising outcomes, the results of experimental data involving human or animal cells cannot be related directly to an in vivo experimental design.
The authors clearly pointed out at the conclusion and discussion that they will carry on in vivo studies exploring the effects of glycyrrhizin on MUC5AC secretion. This denotes that the research team also humbly accepted that the results of the distinct experiment would need in vivo studies for and confirmations of the real therapeutic effect of glycyrrhizin in vivo. Overall, although there are many advantages to this study, the primary disadvantage is that it's an in vitro study which affects the universality of the results for direct application in clinical treatment thus this should be further studied in vivo to prove efficacy.
- Regarding the ERK and p38 pathways targeted by glycyrrhizin in reducing the MUC5AC and MUC5B, further validation is needed. Since the reducing trends of MUC5AC at the dosage of glycyrrhizin are not consistent with those reducing the phosphorylation of both ERK and p38, and combining with the large deviations presented in the dephosphorylation of either ERK or p38 by glycyrrhizin, strongly suggests a possibility that these two molecules might not be the main targets in glycyrrhizin-mediated reduction of MUC5AC.
- The phosphorylations of both ERK and P38 in the CNTL group in Fig 4 are quite different from those in Fig 8 and Fig 9.
4.Line 61, IF-1 beta should be IL-1 beta
Author Response
Comments 1. Limitations of in vitro study: The study was carried out in vitro in the NCI-H292 human airway epithelial cell line. Even though this cell model can be used for mimicking the inflammatory state of human asthmatic lung cells (using PMA stimulation) and has yielded promising outcomes, the results of experimental data involving human or animal cells cannot be related directly to an in vivo experimental design.
The authors clearly pointed out at the conclusion and discussion that they will carry on in vivo studies exploring the effects of glycyrrhizin on MUC5AC secretion. This denotes that the research team also humbly accepted that the results of the distinct experiment would need in vivo studies for and confirmations of the real therapeutic effect of glycyrrhizin in vivo. Overall, although there are many advantages to this study, the primary disadvantage is that it's an in vitro study which affects the universality of the results for direct application in clinical treatment thus this should be further studied in vivo to prove efficacy.
Response 1: Thank you for your helpful comments. We agree with your point that in vivo experiments are necessary. We plan to verify the effects of liquiritin using asthma model mice and other relevant models. A corresponding statement has been added at the end of the Discussion section (line 283-288 and 389-390) to indicate that in vivo experiments will be conducted in future studies.
Comments 2. Regarding the ERK and p38 pathways targeted by glycyrrhizin in reducing the MUC5AC and MUC5B, further validation is needed. Since the reducing trends of MUC5AC at the dosage of glycyrrhizin are not consistent with those reducing the phosphorylation of both ERK and p38, and combining with the large deviations presented in the dephosphorylation of either ERK or p38 by glycyrrhizin, strongly suggests a possibility that these two molecules might not be the main targets in glycyrrhizin-mediated reduction of MUC5AC.
Response 2: We agree that additional molecular pathways may be involved in the function of liquiritin. We are currently planning to investigate the role of JNK, which is known to induce MUC5AC production. A sentence discussing the potential involvement of JNK has been, accordingly, added to the Discussion section (line 277-282).
Comments 3. The phosphorylations of both ERK and P38 in the CNTL group in Fig 4 are quite different from those in Fig 8 and Fig 9.
Response 3: We agree with your point. The phosphorylation levels of both ERK and p38 shown in the CNTL group of Figure 4 were analyzed in PMA-untreated cells and were significantly lower compared with those in PMA-treated cells (Figures 8 and 9). Therefore, the detection in Figure 4 required an extremely long exposure time (approximately 100-fold longer), which accounts for the observed signal differences.
Comments 4. Line 61, IF-1 beta should be IL-1 beta
Response 4: We have made the correction as you pointed out (line 70-73).
Reviewer 3 Report
Comments and Suggestions for Authors
The airway surface in humans is protected by a mucus layer primarily composed of MUC5AC and MUC5B. In asthma, excessive secretion of these mucins, particularly MUC5AC, leads to airway obstruction and worsening symptoms. This manuscript evaluates the inhibitory effects of liquiritin, a flavonoid from licorice, on both intracellular and secreted MUC5AC and MUC5B in human airway epithelial cells (NCI-H292). The results indicate that liquiritin suppresses mucin expression in a dose-dependent manner, potentially through inhibition of the ERK and p38 signaling pathways.
1. The research gap is implied but not clearly articulated as a distinct rationale for the study.
2. The objective statement at the end is verbose and lacks focus.
3. The rationale for chosen liquiritin concentrations is not explicitly justified.
4. The PMA stimulation protocol lacks detail regarding timing and treatment sequence.
5. Text largely repeats data shown in figures without providing interpretive context.
6. Statistical findings are reported, but biological significance is not discussed.
7. Initial paragraphs reiterate results rather than interpreting or contextualizing them.
8. Limitations of the in vitro model are not acknowledged.
9. There is minimal discussion of how findings compare to related studies or existing therapeutics.
10. The concluding remarks are overly general and lack nuance regarding translational relevance.
11. No future research directions are proposed beyond a brief mention of in vivo validation.
Author Response
Comments 1. The research gap is implied but not clearly articulated as a distinct rationale for the study.
Response 1: Thank you for your kind comments. We agree with this comment. Regarding this issue, we have, accordingly, added the description to the Introduction section (line 48-55 and 75-78).
Comments 2. The objective statement at the end is verbose and lacks focus.
Response 2: We agree with this comment. we have, accordingly, modified line 81-83 in the Introduction section and line 283-288 in the Discussion section. And we added Experimental design (line 290-302) in the Materials and Methods section.
Comments 3. The rationale for chosen liquiritin concentrations is not explicitly justified.
Response 3: We agree with this comment. We have, accordingly, modified line 86-90 in the Results section.
Comments 4. The PMA stimulation protocol lacks detail regarding timing and treatment sequence.
Response 4: Regarding this issue, we have, accordingly, modified line 317-323 in the Materials and Methods section.
Comments 5. Text largely repeats data shown in figures without providing interpretive context.
Response 5: Regarding this issue, we have, accordingly, added line 101-102 and 165-167 in the Results section.
Comments 6. Statistical findings are reported, but biological significance is not discussed.
Response 6: Regarding this issue, we have, accordingly, modified line 277-288 in the Discussion section and line 389-390 in the Conclusion section.
Comments 7. Initial paragraphs reiterate results rather than interpreting or contextualizing them.
Response 7: Regarding this issue, we have largely modified line 245-247 in the Discussion section.
Comments 8. Limitations of the in vitro model are not acknowledged.
Response 8: Regarding this issue, we have, accordingly, added line 284-285 in the Discussion section.
Comments 9. There is minimal discussion of how findings compare to related studies or existing therapeutics.
Response 9: Regarding this issue, we have, accordingly, added the description about the effect of Emodin and Pyronaridine in line 259-263 and 267-268 in the Discussion section.
Comments 10. The concluding remarks are overly general and lack nuance regarding translational relevance.
Response 10: Regarding this issue, we have, accordingly, added line 389-390 in the Discussion section.
Comments 11. No future research directions are proposed beyond a brief mention of in vivo validation.
Response 11: Regarding this issue, we have, accordingly, modified line 277-288 in the Discussion section.
Reviewer 4 Report
Comments and Suggestions for Authors
The article "Liquiritin suppresses intracellular and secreted MUC5AC and MUC5B in human airway epithelial cells" presents interesting results that are consistent with the proposed methodology. The introduction is appropriate to the topic, presents up-to-date references, and the discussion is well-written at the pharmacodynamic level. However, I have some suggestions.
1. Why were these concentrations chosen?
2. Standardize the font on lines 235-236.
3. In the Materials and Methods section, a section on the experimental design is missing.
4. Where were the analyzed compounds obtained? How were they dissolved?
5. Are the results presented as the mean +/- standard error or standard deviation?
6. Which post-hoc ANOVA was used: Tukey or DuNet?
7. In the references, include the year of publication in bold and the journal name in italics.
Author Response
Comments 1. Why were these concentrations chosen?
Response 1: Thank you for your helpful comments. Cytotoxicity was observed when the concentration of liquiritin exceeded 400 µM, which is why higher concentrations were not used in this study. High concentration data has been added to Fig1A and 1B. Regarding this, we have, accordingly, modified line 86-90 as description in the Results section.
Comments 2. Standardize the font on lines 235-236.
Response 2: We have made the correction as you pointed out (line 245-247 in the revised manuscript).
Comments 3. In the Materials and Methods section, a section on the experimental design is missing.
Response 3: As suggested, we have added a description of the experimental design to the Materials and Methods section (line 290-302).
Comments 4. Where were the analyzed compounds obtained? How were they dissolved?
Response 4: We have added, accordingly, additional details about the compounds and their dissolution in ethanol in Section 4.2. Cell culture, of the Materials and Methods section (line 312-313).
Comments 5. Are the results presented as the mean +/- standard error or standard deviation?
Response 5: The results are presented as mean ± standard deviation. This has been added in each figure legend accordingly (line 119).
Comments 6. Which post-hoc ANOVA was used: Tukey or DuNet?
Response 6: The Dunnett’s test was used for post-hoc analysis following ANOVA. A description of this has been added, accordingly, to Section 4.7, Statistical Analysis, of the Materials and Methods (line 376).
Comments 7. In the references, include the year of publication in bold and the journal name in italics.
Response 7: We have corrected it as you pointed out (line 408-535).
Round 2
Reviewer 1 Report
Comments and Suggestions for Authors
The authors have revised the manuscript and addressed all of my comments and concerns. I recommend accepting the manuscript for publication.
Reviewer 2 Report
Comments and Suggestions for Authors
The response of author to comment is adequate and appreciated. I have no more questions
Reviewer 3 Report
Comments and Suggestions for Authors
Accept in present form.